



# Exploring Deep Learning for Air Pollutant Emission Estimation

Lin Huang,[★,1] Song Liu,[★,2,3] Zeyuan Yang,[4] Jia Xing,[2,3] Jia Zhang,[1] Jiang Bian,[1] Siwei Li,[5,6] Shovan Kumar Sahu,[2,3] Shuxiao Wang,[2,3] Tie-Yan Liu[1]

[1]Microsoft Research, Beijing, China
[2]State Key Joint Laboratory of Environmental Simulation and Pollution Control, School of Environment, Tsinghua University, Beijing, China
[3]State Environmental Protection Key Laboratory of Sources and Control of Air Pollution Complex, Beijing, China
[4]School of Economics and Management, Tsinghua University, Beijing, China
[5]School of Remote Sensing and Information Engineering, Wuhan University, Wuhan, China
[6]State Key Laboratory of Information Engineering in Surveying, Mapping and Remote Sensing, Wuhan University, Wuhan, China
★These authors contributed equally to this work.

*Correspondence to:* Jia Xing (xingjia@tsinghua.edu.cn); Jia Zhang (zhangjia@microsoft.com)

**Abstract.**   The inaccuracy of anthropogenic emission inventory on a high-resolution scale due to insufficient basic data is one
of the major reasons for the deviation between air quality model and observation results. A bottom-up approach, as a typical emission inventory estimation approach, requires a lot of human labor and material resources, and a top-down approach focuses on individual pollutants that can be measured directly and relies heavily on traditional numerical modelling. Lately, deep neural network has achieved rapid development due to its high efficiency and non-linear expression ability. In this study, we proposed a novel method to model the dual relationship between emission inventory and pollution concentration for emission inventory
estimation. Specifically, we utilized a neural network based comprehensive chemical transport model (NN-CTM) to learn the complex correlation between emission and air pollution. We further updated the emission inventory based on backpropagating the gradient of the loss function measuring the deviation between NN-CTM and observations from surface monitors. We first mimicked the CTM model with neural networks (NN) and achieved a relatively good representation of CTM with similarity reaching 95%. To reduce the gap between CTM and observations, the NN model would suggest an updated emission of $NO_x$,
$NH_3$, $SO_2$, VOC and primary $PM_{2.5}$ which changes by -1.34%, -2.65%, -11.66%, -19.19% and 3.51%, respectively, on average of China. Such ratios of $NO_x$ and $PM_{2.5}$ are even higher (~10%) particularly in Northwest China where suffers from large uncertainties in original emissions. The updated emission inventory can improve model performance and make it closer to observations. The mean absolute error for $NO_2$, $SO_2$, $O_3$ and $PM_{2.5}$ concentrations are reduced significantly by about 10%~20%, indicating the high feasibility of NN-CTM in terms of significantly improving both the accuracy of emission inventory as well
as the performance of air quality model.





## 1 Introduction

The clean air policies have been implemented by China government since 2010 which has been effectively reducing the pollutant concentrations such as sulfur dioxide ($SO_2$), nitrogen oxides ($NO_x$) (Zheng et al., 2018). Nevertheless, China still faces challenges in addressing $O_3$ and $PM_{2.5}$ pollutions. Particularly, the level of ozone ($O_3$) in China has increased by 1.3%

from 2013 to 2017 (Li, 2019); moreover, concentrations of $PM_{2.5}$ (particulate matter with an aerodynamic diameter less than 2.5 μm) in most Chinese cities still far exceed the World Health Organization (WHO) recommended values ($<10$ μgm$^{-3}$), leading to frequent heavy pollution events (Guo et al., 2014; Richter et al., 2005; Vesilind et al., 1988). Such high pollutant concentration may substantially affect human health given air pollution has being ranked fifth in global risk factors for mortality (Institute, 2019).

A prerequisite of effectively controlling air pollution lies in accurate knowledge of the related emission sources. A well-established emission inventory should summarize the amount of pollutants emitted into atmosphere from all sources in a specific region and time span (Institute, 2019). A typical bottom-up approach is adopted to develop the emission inventory through investigation of emission sources in Air Benefit and Cost and Attainment Assessment System Emission Inventory (ABaCAS-EI) (Zheng et al., 2019) and Multi-resolution Emission Inventory (MEIC) (He, 2012) developed by Tsinghua

University, wherein the activity rate of each source is multiplied with emission factor (Vallero and Daniel, 2018). Such technology-oriented bottom-up emission inventory can reflect the types of technology operated in China but has limitation in actual application because of its need for labor power and material resources, especially in cities where it is hard to support thorough investigation (Xing et al., 2020b). What's more, varied assumptions for activity rate and emission factor from different studies result in large uncertainties (Aardenne and Pulles, 2002). Therefore, the development of a method for efficient,

low-cost, and sufficiently accurate grid-emission information is being considered.

The top-down method, as another typical emission inventory estimation approach, can be used to constrain emission estimation by combining observation results from surface monitors and satellite retrievals. Brioude et al. (2012) has estimated emissions of anthropogenic CO, $NO_x$ and $CO_2$ in the Los Angeles Basin using the FLEXPART Lagrangian particle dispersion model based on the top-down method. Recently, Yang et al. (2021) linked the bottom-up MAPLE model with the top-down CGE

model to evaluate deep decarbonization pathways' (DDP) comprehensive impacts in China. However, most of previous studies merely focused on individual pollutants that can be measured directly (Brioude et al., 2012; Xing et al., 2020a; Yang et al., 2021) and relied on traditional numerical modelling.

On the contrary, neural networks (NN), as a more efficient tool, can also model complex nonlinear relations in the atmospheric system thus converting precursor emissions into ambient concentrations. Due to its ability of end-to-end learning, NN can

automatically extract key features of input data and capture the behaviour of target data, thus has been widely used in atmospheric science recently (Fan et al., 2017; Tao et al., 2019; Wen et al., 2019; Xing et al., 2020a; Xing et al., 2020c). For example, many studies (Fan et al., 2017; Tao et al., 2019; Wen et al., 2019) combined recurrent NN (RNN) and convolutional NN (CNN) to capture spatial and temporal features in air pollution related questions since RNN has a strong capability in





mining temporal patterns from time series data (Cho et al., 2014; Chung et al., 2014; Hochreiter and Schmidhuber, 1997) with

certain ability to handle missing values efficiently (Fan et al., 2017) and CNN exhibits potentials in leveraging spatial dependencies, e.g., in meteorological prediction (Krizhevsky et al., 2012). Furthermore, Xing et al. (2020d) applied NN to the surface response model (RSM), thus significantly enhancing the computational efficiency, demonstrating the utility of deep learning approaches for capturing the nonlinearity of atmospheric chemistry and physics. The application of deep learning improves the efficiency of air quality simulation and can quickly provide data support for the formulation of emission control

policies, so as to adapt to the dynamic pollution situation and international situation. But the use of deep NN to estimate emission inventory is more complex compared to traditional machine learning problems, because there is no precise emission observation that can be used as supervision for model training.

To address all these issues, we proposed a novel method based on dual learning (He et al., 2016), which leverages the primal-dual structure of artificial intelligence (AI) tasks to obtain informative feedbacks and regularization signals, thus enhancing

both the learning and inference process. In terms of emission inventory estimation, if we have a precise relationship from emission inventory to pollution concentrations, we can use the pollution concentrations as a constraint to get accurate emission inventory. In particular, we proposed to employ a neural network based chemical transport model (NN-CTM) with a delicately designed architecture, which is *efficient* and *differentiable* compared to chemical transport model (CTM). Furthermore, when a well-trained NN-CTM can accurately reflect the *direct* and *indirect* physical and chemical reactions between emission

inventory and pollutant concentrations, the emission inventory can be updated by backpropagating the gradient of the error between observed and NN-CTM predicted pollutant concentrations. Figure 1 shows the framework of this study.

The method used for this study is described in Section 2. Section 3 takes China emission inventory estimation as an example to demonstrate the superiority of our method. In section 4, we make a conclusion and discuss some possible future work.

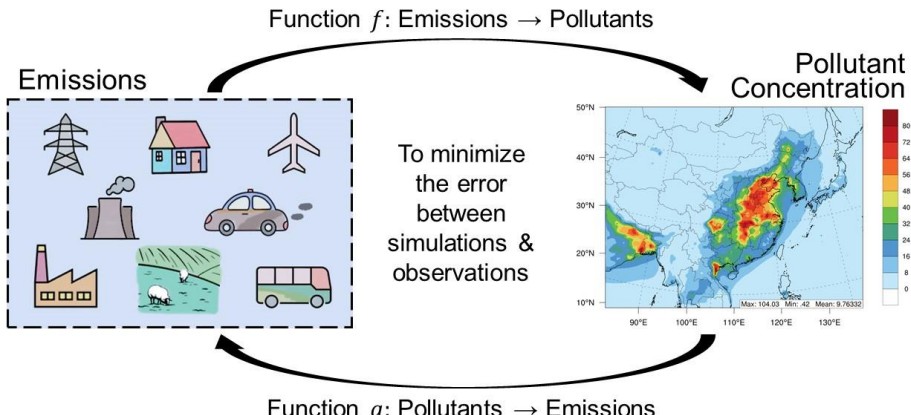


**Figure 1: Framework of this study.**



## 2.Method

### 2.1 Main Framework

The task of emission inventory estimation can be naturally formalized into a typical dual learning framework. Concretely, we denote $x_t$ as the data of emission volumes and meteorological conditions and $y_t$ as the corresponding pollutant concentration at time t. In addition, we denote the mapping function from emission to pollutant concentration as f and that from pollutant concentration to emission as $g$. Since the transformation from emission to pollutant concentration is a continuous process in time, approximately, we have the following equations:

$$y_t = f(x[(t - k + 1):t]),\qquad\qquad(1)$$

$$x_t = g(y[(t - k + 1):t]),\qquad\qquad(2)$$

where $x[i:j]$ is defined as $\{x_i, x_{i+1}, \dots, x_j\}$ for convenience, so is $y[i:j]$.

The formulas above are based on two assumptions:

1.        The pollutant concentration is only dependent on the emission and meteorological conditions in the past $k$ time steps,
e.g., hours or days.

2.        There is a bijection relationship between emission and pollutant concentration. This is a necessary prerequisite for the existence of function $g$.

The first assumption will hold true as long as setting a sufficiently large $k$. The second assumption may not be true unless we introduce more external constraints on emission inventory, since there exists information loss in the process from emission to
pollutant concentration.

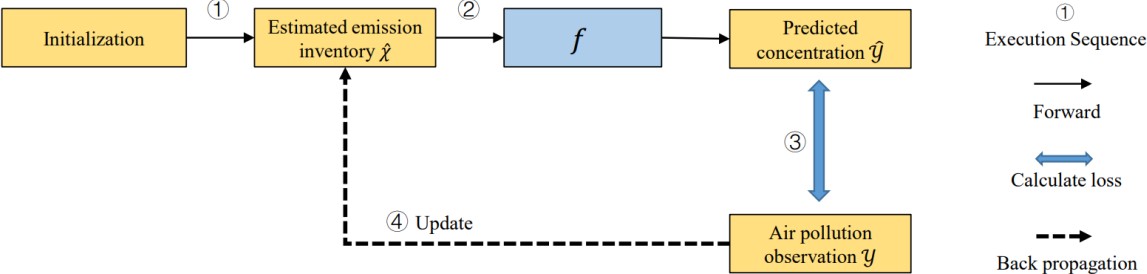

**Figure 2: The whole process of emission inventory estimation.**





In fact, it is quite difficult to learn the function $g$ directly without emission observations as supervision. Hence, we employ dual learning framework to learn function $g$ indirectly through leveraging function $f$. The framework of this process is illustrated in Figure 2. In particular, the whole process of emission inventory estimation includes the following steps:

1.    Use the existing emission inventory which is still not accurate enough as the initial emission data $\hat{\mathcal{X}}$.

2.    Given $\hat{\mathcal{X}}$, calculate the corresponding predicted pollutant concentration data $\hat{\mathcal{Y}}$.

3.    Calculate the loss between the observed values of pollutants $\mathcal{Y}$ and the predicted pollutant concentrations $\hat{\mathcal{Y}}$.

4.    Adjust the estimated emission inventory $\hat{\mathcal{X}}$ by backpropagating the gradient of the loss based on function $f$.

5.    Repeat step 2-4 until achieving sufficient accuracy for predicted concentration.

Although the chemical transport model (CTM) system can handle the transition from emission to pollutant concentration, it is not differentiable, which makes it quite hard to update emission inventory through backpropagation algorithm in the dual

learning framework. In order to establish a differentiable CTM, we propose to build a neural network based chemical transport model (NN-CTM) as the system approximation. More details will be described in the following subsections.

## 2.2 Deep Neural Network based Chemical Transition Model Approximation

Pollutant concentration is usually estimated using CTM which uses emission inventory as input. In the dual learning framework, this input will be updated in turn based on observed concentrations through the backpropagation algorithm. This requires the

CTM be differentiable. To this end, we propose to use deep neural networks to approximate the CTM system. Concretely, to learn this neural network based chemical transport model (NN-CTM), we apply a supervised learning approach that leverages the training data whose input is the same to that of CTM and corresponding label is the output of CTM. The whole architecture is shown in Figure 3.



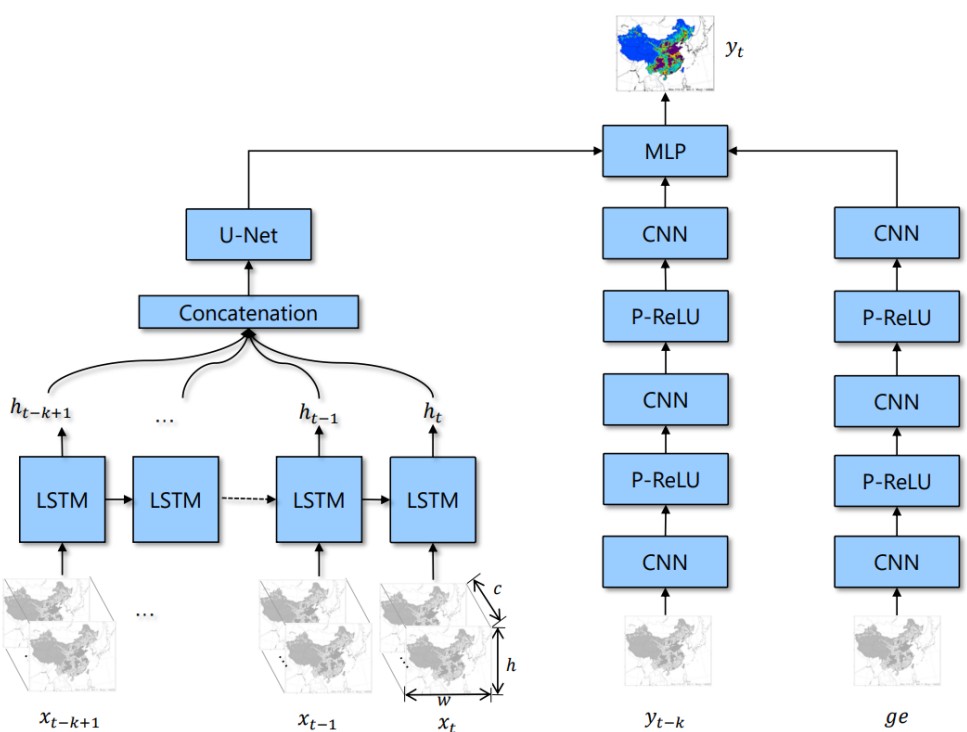


**Figure 3: NN-CTM structure. $c$ represents channel, which consists of emission inventory and meteorological data. $h, w$ represent the height, width of input. $ge$ is geographic information. We employ long short term memory (LSTM) to capture the temporal information, and U-Net to capture the spatial information. CNN represents convolution network. P-ReLU (He et al., 2015b) is a nonlinear activation function. MLP means multiple layers of perceptrons with threshold activation. The model structure is also**
**named as LSTM-U-Net.**

The input data of our NN-CTM are similar to that of CTM, including emission inventory, meteorology and geographical data. The first two are time-dependent data, while the last one, denoted as $ge$, is static data. In the Eulerian grid based CTM system, for each time step $t$, the dynamic input data $x_t$ is a matrix with dimension $w \times h \times c$. The concentration is simulated

continuously in a continuous time sequence. Unlike CTM, the NN-CTM cannot deal with too long data sequence. Thus, we just use the data from past $k$ time steps (i.e, $x[t - k + 1:t]$) as input for the pollutant concentration estimation $y_t$. At the same time, we add $y_{t-k}$ as a supplementary input data into the network. Same as CTM, the output $y_t$ of NN-CTM is a matrix with dimension $w \times h \times l$, where $l$ is the number of concerned pollutant species.

The NN-CTM consists of three branches: two CNN branches for $y_{t-k}$ and $ge$, and one long short term memory (LSTM)

(Hochreiter and Schmidhuber, 1997) with U-Net (Ronneberger et al., 2015) branch. The CNN branches are used to extract features for $y_{t-k}$ and geographical information. We employ parametric rectified linear unit (P-RELU) (He et al., 2015b) as the non-linear activation function in these branches to improve model fitting with nearly zero extra computational cost and little overfitting risk. We adopt the architecture of combining LSTM and U-Net based on the understanding of temporal-spatial relationship in emission inventory. In temporal dimension, pollutants are the accumulation of historical emissions. In spatial

dimension, adjacent grids will affect each other because of meteorological and diffusion factors. The LSTM layer is used to aggregate information from time series data $x[t - k + 1:t]$. The aggregated sequence of hidden states $h_{t-k+1}, \ldots, h_t$ will be concatenated and entered into U-Net block. U-Net is a widely adopted pixel-to-pixel model which can effectively utilize neighbour information. We employ 2-layers U-Net as shown in Figure 4 to capture the spatial information between grids.

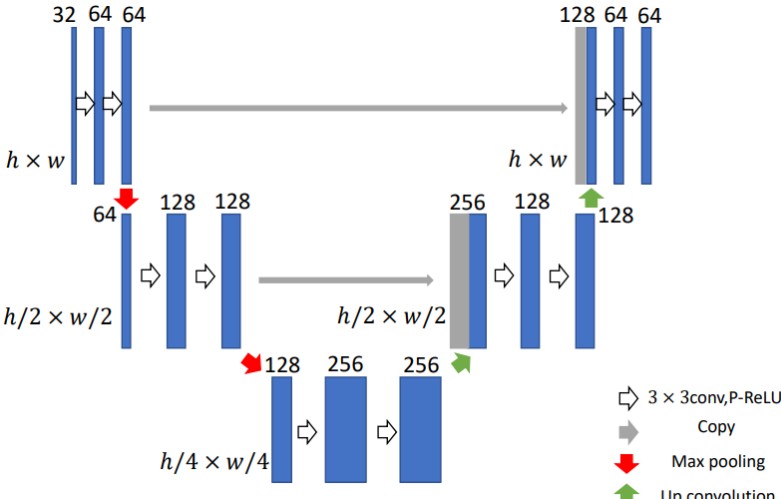


**Figure 4: U-Net structure (2-layers). The model structure yields a u-shaped architecture. 3×3 conv is a convolution (Huang et al., 2016) function. P-ReLU (Huang et al., 2016) is a nonlinear activation function. Max pooling is a down sample function. Up convolution (Zeiler et al., 2010) is a deconvolution function, which is also named as up sample function.**

In the training process, we take $(\mathcal{X}_{CTM}, \mathcal{Y}_{CTM})$ as training dataset, where $\mathcal{X}_{CTM}$ is the input data of the CTM system while $\mathcal{Y}_{CTM}$ is the corresponding output. Since relative changes in pollutant concentrations are the metric often used by policymakers, we adopt an objective function that measures the relative loss between NN-CTM predicted and CTM-simulated pollutant concentrations. We denote the output of NN-CTM as $\hat{\mathcal{Y}}_{NN}$, and have:

$$L\big(\hat{\mathcal{Y}}_{NN}, \mathcal{Y}_{CTM}\big) = \frac{1}{Nhwl} \sum_{n=1}^{N} \sum_{i,j,c} \big(|\hat{y}_{i,j,c}^{(n)} - y_{i,j,c}^{(n)}|\big) \, , \tag{3}$$

$$g_w = \frac{\partial L(\hat{\mathcal{Y}}_{NN}, \mathcal{Y}_{CTM})}{\partial w} \, , \tag{4}$$

where $N$ is the number of samples, $i \in [1, h]$, $j \in [1, w]$ and $c \in [1, l]$, and $y_{i,j,c}^{(n)}$ represents the concentration of pollutant $c$-th in grid with location $(i, j)$ in the $n$-th sample. The parameters of NN-CTM will be updated based on the gradients given by $g_w$, and the Adaptive Moment (Adam) estimation (Kingma and Ba, 2014) is used as optimizer.



## 2.3 Emission Inventory Estimation Based On NN-CTM

Given a well-trained NN-CTM whose approximation accuracy is high enough for predicting pollutant concentrations, the emission inventory can be updated based the error between observed and NN-CTM predicted pollutant concentrations.

In particular, we make the relationship between emission and pollutant concentration more robust by fixing the trained LSTM-U-Net model parameter. Then by training NN-CTM parameter, we adjust the input emission inventory to minimize the loss between NN-CTM output and observation. Such loss can be formally defined as:

$$L\left(\hat{\mathcal{Y}}_{NN}, \mathcal{Y}_{obs}^*\right) = \frac{1}{Nhwl} \sum_{n=1}^{N} \sum_{i,j,c} M_{i,j}\left(|\hat{y}_{i,j,c}^{(n)} - y_{i,j,c}^{*(n)}|\right), \tag{5}$$

$$g_e = \frac{\partial L(\hat{\mathcal{Y}}_{NN}, \mathcal{Y}_{obs}^*)}{\partial e}, \tag{6}$$

where $\mathcal{Y}_{obs}^*$ represents observed pollutant concentration (we will use average value in case of multiple observation stations in a grid), $M_{i,j}$ is a binary indicator variable indicating whether or not there is a site monitoring equipment in grid $(i, j)$. The emission inventory will be updated by backpropagating the gradient $g_e$. The stochastic gradient descent (SGD) method (Bottou,

2010) is used as optimizer.

## 3. Experiments and Results Analysis

In this section, we apply our proposed method for emission inventory estimation in China 2015. In the following, we will first describe the data and CTM configuration. After that, we will show experimental results in terms of the accuracy of NN-CTM. Then, we conduct further analysis on the prior emission inventory and our emission inventory estimation results.

### 3.1 Data and CTM Configuration

The prior emission inventory ABaCAS-EI with high spatial and temporal resolution is based on the bottom-up method, including primary pollutants such as $NO_x$, ammonia ($NH_3$), $SO_2$, volatile organic compounds (VOC) and primary $PM_{2.5}$. ABaCAS-EI is a grid-unit-based emission inventory including sources of power, cement, steel industries, and mobile sources. It also takes into consideration of technical progress and more stringent emission standards (Zheng et al., 2019). The prior

emission inventory is initially used for NN-CTM training and then updated as per the proposed method of dual learning.

Geographical data is a fixed attribute of one grid, like land type, mountains, depressions or elevation, etc. and in this study is obtained from the Moderate Resolution Imaging Spectroradiometer (MODIS) with 15s resolution (Friedl et al., 2002).

Meteorological conditions are simulated from the Weather Research and Forecasting model (WRF, version 3.7). WRF configuration includes Morrison microphysics scheme (Morrison et al., 2009), RRGM radiation scheme (Mlawer et al.,

1998; Mlawer et al., 1997), Pleim-Xiu land surface scheme (Pleim and Xiu, 1995; Xiu and Pleim, 2001), ACM2 planetary boundary layer (PBL) physics scheme (Pleim, 2007) and Kain-Fritsch cumulus cloud parameterization (Kain,




2004), which matches our previous studies (Ghil and Malanotte-Rizzoli, 1991; Wikle, 2003). Data assimilation is adopted in WRF simulations based on observation data for the upper air and surface from National Centers for Environmental Prediction (NCEP) datasets. The simulated temperature, humidity, wind speed and direction has good agreement with the observations from the National Climatic Data Center (NCDC, https://www.ncdc.noaa.gov/data-access/land-based-station-data/) (Ding et al., 2019; Liu et al., 2019; Zhao et al., 2013).

The Community Multiscale Air Quality Model (CMAQ, version 5.2) configured with the AERO6 aerosol module (Appel et al., 2013) and the Carbon Bond 6 (CB6) gas-phase chemical mechanism (Sarwar et al., 2008) is chosen as the representative CTM to simulate pollutant concentrations (Appel et al., 2018; Byun, 1999). Hourly observation data for air pollution (including SO$_2$, NO$_2$, O$_3$ and PM$_{2.5}$) is obtained from the China National Environmental Monitoring Centre (http://beijingair.sinaapp.com/), which are used for adjusting emission inventory.

The simulation domain covers mainland China and portions of surrounding countries with a 27km × 27km horizontal resolution (with $h = 182$ and $w = 232$) and 14 vertical layers from ground to 100 hPa. Simulations are performed in January, April, July, and October 2015 to represent winter, spring, summer, and autumn, respectively. A 5-day simulation spin-up was performed to minimize the effects of initial conditions. Pollutant concentrations are analysed at monthly averages.

### 3.2 NN-CTM Learning and Evaluation

*Training parameters*. The parameter of NN-CTM was optimized using Adam optimizer with a mini-batch size of 8. A learning rate of 0.001 was used. To reduce the risk of over-fitting, we applied weight regularization on all trainable parameters during training and fine-tuning. The NN-CTM was trained for 30000 epochs.

*Metrics*. Model performance was evaluated using mean absolute error (MAE) calculated using the following equation:

$$L(\hat{\mathcal{Y}}_{NN}, \mathcal{Y}_{CTM}) = \frac{1}{Nhwl}\sum_{n,i,j,c}(|\hat{y}_{i,j,c}^{(n)} - y_{i,j,c}^{(n)}|), \tag{7}$$

Where $N, h, w, l$ are the number of samples, height, width and the number of observed pollutants in each grid, respectively, further $n \in [1, N]$, $i \in [1, h]$, $j \in [1, w]$ and $c \in [1, l]$.

*Evaluation*. We examined the performance of NN-CTM to check whether it has learnt the relationship between emission and pollutant concentration.

We trained NN-CTM on the data of first 22 days in January, April, July and October 2015 and tested it on the remaining successive 8 days of each month. As listed in Table 1, NN-CTM (with LSTM-U-Net) can well reproduce the spatial and temporal relation with a small MAE of 0.27, 0.17, 1.39 ppbv and 1.46 µg m$^{-3}$ for NO$_2$, SO$_2$, O$_3$ and PM$_{2.5}$, respectively, on average of four months. Results suggest that the NN-CTM can well reproduce the CTM within an acceptable bias, thus can be used for emission adjustment. Such bias (<4%) is much smaller than that of simulation compared to observations which is normally more than 10% even 20%.





**Table 1: Evaluation of NN-CTM simulation in China (mean absolute error between CTM and NN-CTM). LSTM-U-Net is our proposed method. And then, to compare the model performance, we select another professional deep neural network method residual network (ResNet) (He et al., 2015a).**

| Model | NN-CTM (with LSTM-U-Net) | | | | NN-CTM (with ResNet) | | | |
|---|---|---|---|---|---|---|---|---|
| Variables | $PM_{2.5}$ ($\mu g\ m^{-3}$) | $O_3$ (ppbv) | $NO_2$ (ppbv) | $SO_2$ (ppbv) | $PM_{2.5}$ ($\mu g\ m^{-3}$) | $O_3$ (ppbv) | $NO_2$ (ppbv) | $SO_2$ (ppbv) |
| Jan. | 1.65 | 1.39 | 0.34 | 0.25 | 1.65 | 1.44 | 0.36 | 0.26 |
| Ari. | 1.74 | 1.46 | 0.25 | 0.16 | 1.73 | 1.64 | 0.26 | 0.18 |
| Jul. | 1.04 | 1.38 | 0.23 | 0.12 | 1 | 1.45 | 0.25 | 0.13 |
| Oct. | 1.43 | 1.34 | 0.27 | 0.16 | 1.53 | 1.44 | 0.29 | 0.17 |
| Average | 1.46 | 1.39 | 0.27 | 0.17 | 1.48 | 1.49 | 0.29 | 0.19 |
| Error (Unit: %) | 3.6 | 3.9 | 1.9 | 2.2 | 3.7 | 4.3 | 2.1 | 2.5 |

In order to further verify the superiority of our model architecture, we employed the ResNet (He et al., 2015a), another widely adopted deep NN method in image processing. Compared to ResNet, the performance of NN-CTM (with LSTM-U-Net) was superior, with improved MAE of 0.02, 0.02, 0.10 ppbv and 0.02 $\mu g\ m^{-3}$ for $NO_2$, $SO_2$, $O_3$ and $PM_{2.5}$, respectively, on average of four months, as listed in Table 1.

### 3.3 Emission Inventory Updating and Analysis

A well trained NN-CTM is used to update the emission inventory through back propagation using stochastic gradient descent SGD (Bottou, 2010) optimizer with a mini-batch size of 2. The learning rate is 0.1. The optimization of emissions is achieved after 10000 epochs.



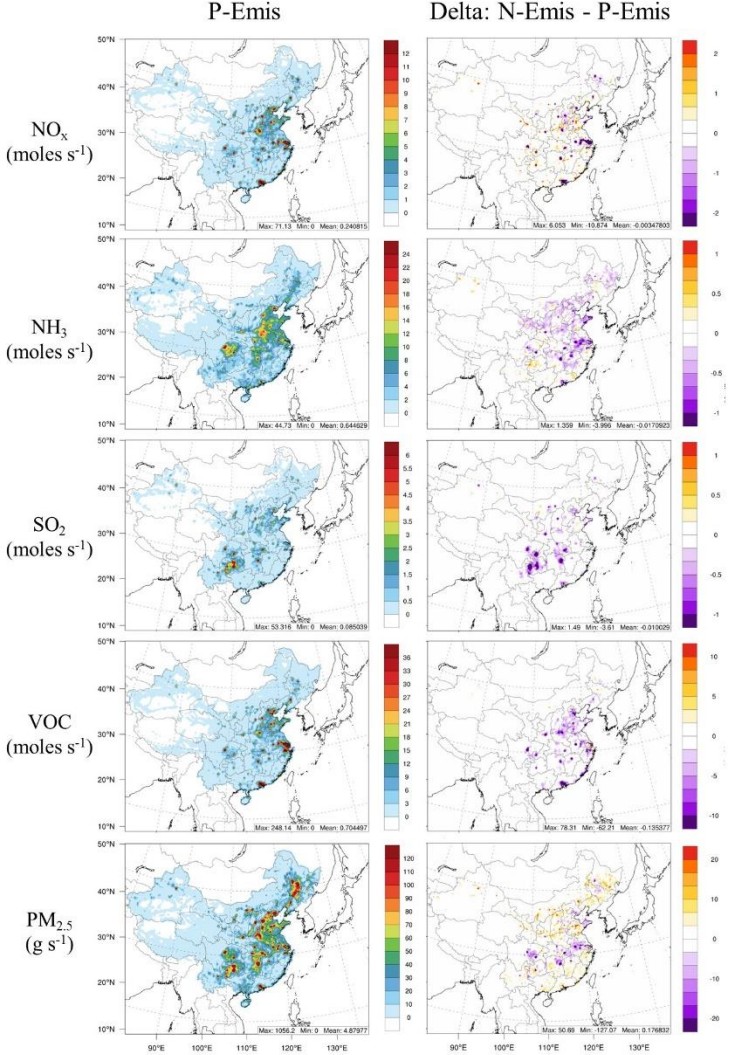

**Figure 5: Emission rates of NO$_x$, NH$_3$, SO$_2$, VOC and primary PM$_{2.5}$ in P-Emis and their changes in N-Emis.**

For convenience, we denote the emission inventory from ABaCAS-EI as prior emissions (P-Emis) and the updated emission

inventory as NN-emission (N-Emis), which is constrained by station observations. Compared with P-Emis, N-Emis has

adjusted emission rates of NO$_x$, NH$_3$, SO$_2$, VOC and primary PM$_{2.5}$ as per the difference between simulated concentrations

and the observed values of pollutants in each grid, as shown in Figure 5. Average emission rates of NH$_3$, SO$_2$ and VOC in

most grids tend to decrease, while that of primary PM$_{2.5}$ tend to increase except for in the Yangtze River Basin, which may be

related to the non-included dust emission. Changes in emission rate of NO$_x$ vary a lot by regions, and such changes are

concentrated in urban areas. The distribution of N-Emis for each grid is consistent with P-Emis, indicating that the deep





learning method in this study can identify the distribution of emission sources and focus on the calibration in high-emission areas.

Annual anthropogenic emissions in China for $NO_x$, $NH_3$, $SO_2$, VOC and primary $PM_{2.5}$ in P-Emis are 20.44, 10.39, 14.40, 23.05 and 7.19 Mt, respectively (Liu et al., 2020), while in N-Emis changed by -1.34%, -2.65%, -11.66%, -19.19% and 3.51%, respectively.

**Table 2: Change ratios of N-Emis compared with P-Emis in four months. Unit: %.**

| Month | Variables | | | | |
|---|---|---|---|---|---|
| | $NO_x$ | $NH_3$ | $SO_2$ | VOC | $PM_{2.5}$ |
| Jan. | 3.72 | 1.88 | -12.38 | -25.36 | 4.64 |
| Apr. | -1.49 | -2.56 | -8.96 | -18.27 | 4.69 |
| Jul. | -11.68 | -2.29 | -11.42 | -12.8 | 1.8 |
| Oct. | 3.6 | -4.61 | -13.32 | -19.03 | 2.4 |
| Average | -1.34 | -2.65 | -11.66 | -19.19 | 3.51 |

The sensitivity of change ratios to different seasons varies. Table 2 lists the change ratios of N-Emis compared to P-Emis in four months. As for N-Emis, $NO_x$ increases in January and October by about 3.5~4.0%, while it decreases by more than 10% in July. Emission of $NH_3$ increases in January while decreases in other three months with the highest decrease registered in October. Emission of $SO_2$ tends to decrease in all four months with ratios around 10%. Emission of VOC also tends to decrease but with a larger magnitude of about 20% compared to $SO_2$, which may be related to the overestimation of $O_3$. Emission of primary $PM_{2.5}$ tends to increase by less than 5% in four months.

Such changes in emissions are based on mathematical algorithms and thus cannot be explained by physical and chemical processes. The NN method tries to give a solution to make simulation results of all pollutant species closer to observations by compensating the errors in emission inventory. For example, concentrations of $PM_{2.5}$ obtained using P-Emis are generally lower than the observed level, so the emission of primary $PM_{2.5}$ will be increased during the adjustment. $SO_2$ tends to be overestimated using P-Emis, so the adjustment tends to decrease. However, because sulfate is an important component of $PM_{2.5}$, the adjustment of $SO_2$ will be restricted by the underestimation of $PM_{2.5}$. Concentrations of $O_3$ obtained using P-Emis are generally higher than the observed level, so it tends to reduce the emissions of $NO_x$ and VOC, which are precursors of $O_3$, during the adjustment. It is worth noting that the adjustment range of $NO_x$ is much lower than of VOC, because only the observed concentration of $NO_2$ is used as a constraint. Such results are consistent with our previous study (Xing et al., 2020a). In order to further analyse the change of emissions at a regional level, we calculated the four-month average emissions of P-Emis and change ratios of N-Emis for five emission species in Beijing-Tianjin-Hebei region (BTH), the Yangtze River Delta (YRD), the Pearl River Delta (PRD), the Sichuan Basin (SCH) and northwest China (NWC), as highlighted in Figure 6. The





first four areas were selected because they are the main population clusters, and NWC was selected because there are so few observation sites in this area that the constraints are relatively insufficient.

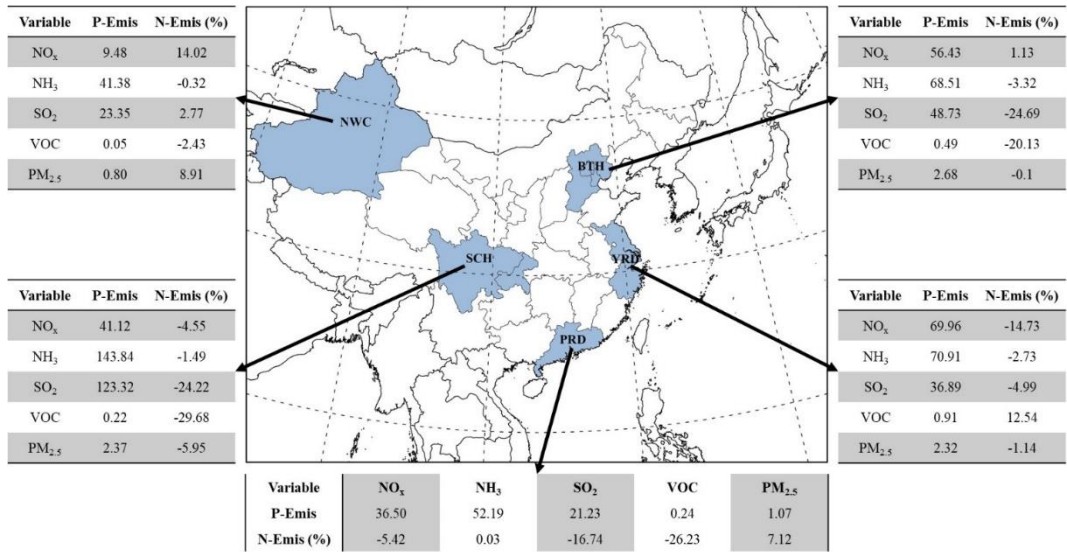

**Figure 6: Five typical regions of China : Beijing-Tianjin-Hebei region (denoted as BTH), the Yangtze River Delta (denoted as YRD, covering Jiangsu, Zhejiang and Shanghai), the Pearl River Delta (denoted as PRD, covering Guangdong), the Sichuan Basin (denoted as SCH, covering Sichuan and Chongqing) and northwest China (denoted as NWC, covering Xinjiang), and their month-average emissions of four months in P-Emis (with unit of kt except for VOC with Mmol) and change ratios in N-Emis (with unit of %).**

The adjustment of emission varies greatly by seasons and regions. Seasonal details are listed in Table 3. The four-month average changes of N-Emis in BTH are highest for $SO_2$ and VOC emissions reaching about -20% while that of $NO_x$, $NH_3$ and primary $PM_{2.5}$ vary by less than 5%. In YRD, NOx and VOC emissions record the highest extent of changes with -14.73% for $NO_x$ and 12.54% for VOC. The range of changes in other emission species is less than 5% (all decrease). Emission of primary $PM_{2.5}$ in PRD increases by about 7%, which is the largest change ratio among four urban regions. Emission of $NH_3$ in PRD changes the least compared with other regions. In SCH, emissions of $SO_2$ and VOC decrease the most (change ratio) compared with other emission species (>20%). Emission of primary $PM_{2.5}$ in SCH, which decreases by 5.95%, shows an opposite trend to that in PRD. As for in NWC, emissions of $NH_3$ and VOC show a small decrease (<5%), while emissions of $NO_x$ and primary $PM_{2.5}$ have a large percentage increase compared with other regions (10%), thus particularly indicating the large inaccuracy in emission inventory in NWC.

**Table 3: Emissions and change ratios in five typical regions of four months.**

| Month | Variables | Version | Regions | | | | |
|---|---|---|---|---|---|---|---|
| | | | BTH | YRD | PRD | SCH | NWC |





| | | | | | | | |
|---|---|---|---|---|---|---|---|
| Jan. | $NO_x$ | P-Emis (kt) | 68.05 | 70.56 | 37.07 | 43.29 | 10.26 |
| | | N-Emis (%) | -7.19 | 6.24 | 4.55 | 2.65 | 8.74 |
| | $NH_3$ | P-Emis (kt) | 28.65 | 24.42 | 6.5 | 24.84 | 5.52 |
| | | N-Emis (%) | 0.67 | 0.59 | 8.3 | 5.13 | 8.64 |
| | $SO_2$ | P-Emis (kt) | 90.13 | 40.16 | 21.72 | 150.67 | 34.06 |
| | | N-Emis (%) | -11.93 | -11.38 | -13.92 | -26.14 | -1.5 |
| | VOC | P-Emis (Mmol) | 0.81 | 0.99 | 0.25 | 0.28 | 0.05 |
| | | N-Emis (%) | -5.53 | -12.73 | -37.52 | -36.39 | 2.61 |
| | $PM_{2.5}$ | P-Emis (kt) | 4.66 | 2.22 | 1.14 | 3.6 | 0.85 |
| | | N-Emis (%) | 1.27 | 10.14 | 15.59 | -0.8 | 9.61 |
| Apr. | $NO_x$ | P-Emis (kt) | 52.43 | 67.17 | 35.21 | 39.05 | 8.72 |
| | | N-Emis (%) | 8.93 | -15.05 | -5.22 | -7.8 | 14.59 |
| | $NH_3$ | P-Emis (kt) | 85.56 | 90.34 | 70.52 | 192.37 | 56.06 |
| | | N-Emis (%) | -3.63 | -2.6 | -0.18 | -1.78 | -0.43 |
| | $SO_2$ | P-Emis (kt) | 33.93 | 34.44 | 20.43 | 110.74 | 20.2 |
| | | N-Emis (%) | -28.14 | -0.92 | -14.17 | -22.97 | 6.77 |
| | VOC | P-Emis (Mmol) | 0.38 | 0.85 | 0.23 | 0.19 | 0.05 |
| | | N-Emis (%) | -28.87 | 13.71 | -25.61 | -29.06 | -2.29 |
| | $PM_{2.5}$ | P-Emis (kt) | 1.81 | 1.96 | 0.94 | 1.79 | 0.62 |
| | | N-Emis (%) | -5.12 | -3.93 | 4.34 | -8.89 | 9.54 |
| Jul. | $NO_x$ | P-Emis (kt) | 50.51 | 72.03 | 36.61 | 41.35 | 9.01 |
| | | N-Emis (%) | -10.62 | -29.84 | -11.46 | -11.94 | 6.86 |
| | $NH_3$ | P-Emis (kt) | 108.78 | 114.78 | 89.82 | 245.68 | 71.16 |
| | | N-Emis (%) | -5.41 | -2.7 | -1.1 | -0.9 | 0.08 |
| | $SO_2$ | P-Emis (kt) | 35.65 | 36.95 | 21.26 | 115.74 | 17.51 |
| | | N-Emis (%) | -38.45 | -4.18 | -19.26 | -19.71 | 12.71 |
| | VOC | P-Emis (Mmol) | 0.39 | 0.92 | 0.24 | 0.21 | 0.05 |
| | | N-Emis (%) | -22.87 | 23.85 | -21.29 | -16.47 | 8.85 |
| | $PM_{2.5}$ | P-Emis (kt) | 2.34 | 3.18 | 1.06 | 2.27 | 0.72 |
| | | N-Emis (%) | 0.88 | -4.94 | 4.17 | -5.4 | 6.91 |
| Oct. | $NO_x$ | P-Emis (kt) | 54.83 | 70.11 | 37.11 | 40.84 | 9.96 |
| | | N-Emis (%) | 14.56 | -19.99 | -9.62 | -1.5 | 25.41 |
| | $NH_3$ | P-Emis (kt) | 50.43 | 53.4 | 41.24 | 110.75 | 32.28 |
| | | N-Emis (%) | -0.53 | -4.52 | 1.54 | -3.79 | -2.54 |
| | $SO_2$ | P-Emis (kt) | 35.68 | 36.08 | 21.52 | 116.47 | 21.74 |
| | | N-Emis (%) | -39.8 | -2.73 | -19.63 | -27.43 | -2.39 |
| | VOC | P-Emis (Mmol) | 0.4 | 0.89 | 0.25 | 0.19 | 0.06 |
| | | N-Emis (%) | -38.33 | 27.98 | -20.39 | -34.42 | -16.41 |
| | $PM_{2.5}$ | P-Emis (kt) | 1.94 | 1.95 | 1.15 | 1.85 | 1.04 |



| | | | | | |
|---|---|---|---|---|---|
| N-Emis (%) | 0.26 | -4.86 | 3.8 | -13.68 | 9.32 |

### 3.4 Accuracy Improvements of CTM Simulation for pollutants with N-Emis

We use the CTM to evaluate the accuracy of P-Emis and N-Emis. The configuration of CTM keeps constant.

Generally, simulations using P-Emis tend to underestimate the $PM_{2.5}$ concentrations and overestimate the $O_3$ concentrations on average of four months in China, which are consistent with our previous studies (Ding et al., 2019; Liu et al., 2019). The underestimation of $PM_{2.5}$ using P-Emis usually appears in Northern and South eastern China, and sometimes occurs in some provinces of the Yangtze River Basin. The simulations of $O_3$ using P-Emis are generally overestimated at observation sites.

Such errors can be narrowed when using N-Emis. We calculated the MAE for each simulation to compare the performances considering all observation sites. After using adjusted emissions (i.e., N-Emis), the MAE for $NO_2$, $SO_2$, $O_3$ and $PM_{2.5}$ concentrations reduced significantly from 7.39 to 5.91 (20.03%), 3.64 to 3.22 (11.54%), 14.33 to 11.56 (19.33%) ppbv and 18.94 to 16.67 (11.99%) $\mu g\,m^{-3}$, respectively, average for total 612 observation stations, as shown in Figure 7. Such improvements prove the advantages of using N-Emis compared with P-Emis. Spatial distributions of comparison between

simulations and observations in 612 sites can be found in Figure 8.

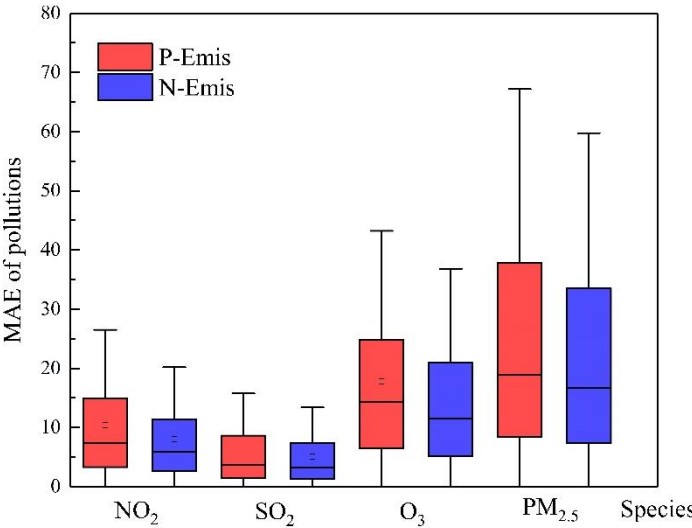

**Figure 7. The MAE of NO₂ (ppbv), SO₂ (ppbv), O₃ (ppbv) and PM₂.₅ (μg m⁻³) concentrations based on P-/N-Emis.**





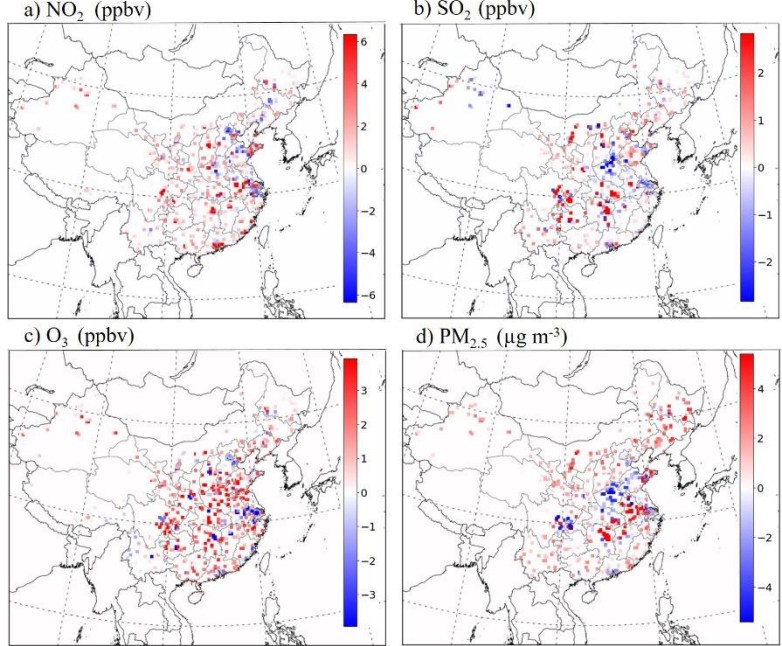

**Figure 8: The MAE of NO₂ (ppbv), SO₂ (ppbv), O₃ (ppbv) and PM₂.₅ (µg m⁻³) concentrations based on P-/N-Emis.**

Difference of monthly simulations using N-emis & P-Emis as input can be used to estimate the seasonal impacts of emission changes. Concentrations of O₃ and PM₂.₅ tend to increase in July while decrease in other months on average for China. Concentrations of NO₂ and SO₂ tend to decrease in four months, which are consistent with the direct trend of emission adjustments.

We also calculated the average concentrations of four pollutants in five typical regions to quantify the degree of improvement in pollutant concentrations after adjusting the emission inventory, as listed in Table 4. Changes in NO₂ and SO₂ concentrations are consistent with adjustments in emissions but are more sensitive, i.e. a small change (~10%) in emission results in a larger proportional change (~20%) in concentration. The reduced SO₂ emissions is an important reason for the improvement of PM₂.₅ overestimations in the Yangtze River Basin. PM₂.₅ concentrations in NWC shows the highest increase (15%) compared with other regions. As the emission inventory in NWC has great potential for improvement (subject to production methods and the acquisition of basic data), the qualitative changes in PM₂.₅ concentrations brought about using NN method seems meaningful. The increase and decrease of NOₓ and VOC emissions directly control the variance in O₃ concentration. Effect of using N-Emis on O₃ concentration is not obvious, with change range of less than 5% in typical regions. Although the adjustment ratio of emissions of O₃ precursors is considerable, the O₃ concentration doesn't change by much. The same can be linked to the complex relationship of precursor emissions of NOₓ and VOC which might not change simultaneously and in the same direction (e.g. increase NOₓ and decrease in VOC or vice-versa) thus resulting in only slight change in O₃ concentration.





**Table 4: Four-month average concentrations of $NO_2$, $SO_2$, $O_3$ and $PM_{2.5}$ in five typical regions using different emission inventories.**

| Variables | Version | Regions | | | | |
|---|---|---|---|---|---|---|
| | | BTH | YRD | PRD | SCH | NWC |
| $NO_2$ (ppbv) | P-Emis | 15.69 | 13.31 | 6.25 | 4.82 | 0.31 |
| | N-Emis | 11.85 | 10.79 | 5.29 | 4.45 | 0.33 |
| $SO_2$ (ppbv) | P-Emis | 6.97 | 4.32 | 1.89 | 4.88 | 0.26 |
| | N-Emis | 5.77 | 3.95 | 1.67 | 3.2 | 0.37 |
| $O_3$ (ppbv) | P-Emis | 34.79 | 41.63 | 40.16 | 41.94 | 41.42 |
| | N-Emis | 35.51 | 39.7 | 38.43 | 40.06 | 41.47 |
| $PM_{2.5}$ ($\mu g\ m^{-3}$) | P-Emis | 46.28 | 44.29 | 22.6 | 25.96 | 2.02 |
| | N-Emis | 45.28 | 41.66 | 22.08 | 23.71 | 2.33 |

## 4. Conclusion and Discussion

In this study, we pioneer the use of machine learning to re-formulate the problem of emission inventory estimation. It creates a new perspective that the data-driven approach can be applied to automatically improve the quality of the emission inventory,

avoiding manual intervention and empirical error. We proposed a differential neural network based chemical transport model (NN-CTM), which achieve a relatively good representation of CTM. And then, we employed backpropagation algorithm to update the emission inventory based on the deviation between observed and NN-CTM predicted pollutant concentrations. In terms of method, we have proposed a novel emission inventory estimation method based on dual learning which consists of dual-loop of emission-to-pollution and pollution-to-emission. Results indicate that our NN based method with adjusted

emission inventory performed better than using prior emissions.

Compared with previous studies, our framework employs dual learning mechanism where in the simulated concentrations are compared to ground observation and the gradient is back propagated to update the emission inventory in each epoch. Results show that new emissions after the adjustment can improve the model performance in simulating the concentrations close to observations. The mean absolute error for $NO_2$, $SO_2$, $O_3$ and $PM_{2.5}$ concentrations reduced significantly by 10% to 20%. This

application uses a constant biogenic emission inventory, so the potential errors in biogenic emissions are also included in the learning of anthropogenic emissions.

Our method can be naturally extended to other fundamental problems, such as $CO_2$ and other greenhouse gas emission inventory estimation, and has broad application prospects, such as building a real-time emission monitoring system based on real-time pollutant observation data.



**Code/Data availability**

The codes for machine learning are available in https://doi.org/10.5281/zenodo.4607127 (Huang et al., 2021), including demo case for this study with input data from Ding et al. (2016) and the China National Environmental Monitoring Centre (http://beijingair.sinaapp.com/). CMAQv5.2 is an open-source and publicly available model developed by the United States Environmental Protection Agency, which can be downloaded at https://doi:10.5281/zenodo.1167892 (Appel et al., 2018).

**Author Contribution**

Lin Huang and Song Liu conceived the research project; Zeyuan Yang analyzed the data; Jia Xing, Jia Zhang, Jiang Bian, Siwei Li, Shuxiao Wang and Tie-Yan Liu provided valuable discussions on research and paper organization; Lin Huang, Song Liu, Zeyuan Yang, Shovan Kumar Sahu, Jia Xing, Jia Zhang and Jiang Bian wrote the paper with contributions from all the authors.

**Competing interests**

The authors declare that they have no conflict of interest.

**Acknowledgement**

This work was supported in part by the National Natural Science Foundation of China (41907190, 51861135102) and the National Key R&D program of China (2017YFC0213005). This work was completed on the "Explorer 100" cluster system of Tsinghua National Laboratory for Information Science and Technology.





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
