# Peer review of "Exploring Deep Learning for Air Pollutant Emission Estimation"

_Geoscientific Model Development, 2021_

## Author Comment (AC1)

We appreciate the reviewer's valuable comments and constructive suggestions which help us improve the quality of the manuscript. We have carefully revised the manuscript according to these comments. Point-to-point responses are provided in the attachment.

The reviewers' comments are in black, our responses are in blue, and the quotes from our manuscript are in italics.

**Reviewer #1**

[Comment]: How does the authors ensure the robustness of the model?

[Response]: We thank the reviewer for the valuable comment. We ensure the robustness of the model from three aspects:

a)  Model structure. Inspired by computer vision tasks, we adopt the batch- normalization (Ioffe and Szegedy, 2015), dropout (Srivastava et al., 2014), L2

regularization (Zhang et al., 2016) to improve the generalization and robustness.

b)  Early stop. When we train the NN-CTM, we split the data into train dataset and validation dataset. As introduced in Sec. 3.1, we trained NN-CTM on the data of the first 22 days in January, April, July, and October 2015 and tested it on the remaining successive 8 days of each month. We stop the model training when the evaluation in validation dataset does not improve within 1000 iterations.

c)  Data augmentation. During training, we employ the noise injection, random rescaling, random rotation method to avoid the overfitting in training dataset.

We have clarified the model robustness in the revised manuscript, as follows:

(Section 2.2, Paragraph 5) *"Model robustness. We ensure the robustness of the model*

*from three aspects: 1) Model structure. Inspired by computer vision tasks, we adopt the*

*batch-normalization (Ioffe and Szegedy, 2015), dropout (Srivastava et al., 2014), L2*

*regularization (Zhang et al., 2016) to improve the generalization and robustness. 2)*

*Early stop. When we train the NN-CTM, we split the data into train dataset and*

*validation dataset, and we stop the model training when the evaluation in validation*

*dataset does not improve within 1000 iterations. 3) Data augmentation. During*

*training, we employ the noise injection, random rescaling, random rotation method to*

*avoid the overfitting in training dataset."*

Reference:

Ioffe S, Szegedy C. Batch Normalization: Accelerating Deep Network Training by

Reducing Internal Covariate Shift. JMLR.org 2015.

Srivastava N, Hinton G, Krizhevsky A, Sutskever I, Salakhutdinov R. Dropout: A

Simple Way to Prevent Neural Networks from Overfitting. Journal of Machine

Learning Research 2014; 15: 1929-1958.

Zhang C, Be Ngio S, Hardt M, Recht B, Vinyals O. Understanding deep learning requires rethinking generalization, 2016.

[Comment]: The authors use the observation data to update the emissions, however they do not mention what happens in case more than one observation station is in the grid.

27 km × 27 km is a large grid size and hence would include many observation stations in one grid. The averaged observed concentration of all stations if used won't serve the purpose to accurately update the emissions at a station.

[Response]: We thank the reviewer for the valuable comment. As mentioned in Section

2.3, we use the average value in case of multiple observation stations in a 27 km $\times$ 27

km grid. We use the same processing method for observations when calculating MAE.

We focus on the emission estimation in one grid, which will be limited by the grid size.

If we want to get the higher resolution emission inventory estimation result (such as focus on one typical region instead of whole China domain), we should use a finer- grained emission inventory as the input. What's more, the lack of observation data in some regions limits our updating, so we are more concerned about making good use of existing observation data.

[Comment]: The entire premise of the model depends on availability of observation data, what happens if data is very sparsely available e.g. say out of 4 neighboring grids only one has observation data how are the emissions in other 3 grids updated?

[Response]: We thank the reviewer for the valuable comment. When we train the NN-

CTM, the long short term memory (LSTM) block is employed to capture the temporal information, and the convolution (U-Net) is employed to capture the spatial information (e.g. the emission inventory, meteorological information, and geographic information of its neighbor grid). That is to say, in NN-CTM, the convolution neural network will capture the surrounding grids' information within the receptive field, and we make a detailed introduction about the receptive field in the answer of next comment which represents the transmission between different grids. Therefore, as shown in Fig. 1, if only the red gird has observation data, the surrounding blue grids' emission inventory within the receptive field will also be updated. At the same time, the grids with a longer
distance will have a lower update weight. In extreme circumstances, if we have no
observation data, our method will not work as we have no more information to adjust
the emission inventory. If the observation data is denser, the emission inventory
estimation is more accurate as it can consider more observation data.

[Figure]

Figure 1: The visualization of neighbor emission update.

We have clarified the relation between observation data and emission inventory in the
revised manuscript, as follows:
(Section 2.3, Paragraph 1) *"The observation data will help update the surrounding*
*grids' emission inventory within the receptive field. However, in extreme circumstances,*
*if we have no observation data, our method will not work as we have no more*
*information to adjust the emission inventory. If the observation data is denser, the*
*emission inventory estimation is more accurate as it can consider more observation*
*data."*

[Comment]: Does the deep learning process consider the impact of transmission
between different grids? The authors are suggested to explain this point in detail.
[Response]: We thank the reviewer for the valuable suggestion to improve the quality
of the paper. The deep learning process has considered the impact of transmission
between different grids. The NN-CTM, which refers to U-Net branch in particular,
employs the convolution neural network to utilize neighbor information effectively. We
visualize a demo case of $3 \times 3$ convolution and $5 \times 5$ convolution in Fig. 2. In U-Net, the stacked of convolution can get the neighbor information with a bigger receptive field (e.g. stacking 5×5 convolution and 5×5 convolution can get a 9×9 convolution), the non-linear function (P-RELU) is employed to improve model fitting with nearly zero extra computational cost and little overfitting risk, and the batch normalization and dropout are employed to enhance the robustness of the model. We calculate that the receptive field of our model is 38×38 grid. In other words, the predicted pollutant concentration is related to its surrounding 38×38 grid's information, which represents the transmission between different grids. Meanwhile, the closer the distance, the greater the contribution.

[Figure]

Figure 2: The visualization of convolution neural network (left: 3*3 kernel size, right: 5*5 kernel size).

We have clarified the impact of transmission between different grids in the revised manuscript, as follows:

(Section 2.2, Paragraph 3) *"In U-Net, the stacked of convolution can get the neighbor information with a bigger receptive field (e.g. stacking 5×5 convolution and 5×5 convolution can get a 9×9 convolution), the non-linear function (P-RELU) is employed to improve model fitting with nearly zero extra computational cost and little overfitting risk, and the batch normalization and dropout are employed to enhance the robustness of the model. We calculate that the receptive field of our model is 38×38 grid. In other words, the predicted pollutant concentration is related to its surrounding 38×38 grid's information, which represents the transmission between different grids. Meanwhile, the closer the distance, the greater the contribution."*

[Comment]: Lines 24-26, Abstract. Please be specific on the simulation year and the emission inventory you applied.

[Response]: We apologize for missing this information, and we have added year 2015 in Abstract.

[Comment]: Line 310, Page 15. I suggest the authors add more description for Figure 8, such as explaining why the performance of using the new emission inventory worsened at some sites.

[Response]: We appreciate the reviewer for the valuable suggestion. We have added more explanations accordingly as follows:

(Section 3.4, Paragraph 2) *"The model performance of most stations has been improved,*

*and a small number of stations with worsen performance show the link between*

*compound pollutants. For example, stations with larger deviations between $PM_{2.5}$*

*simulation results and observations tend to have greatly improved $O_3$ performance, and*

*vice versa."*

[Comment]: The language of the manuscript needs to be further polished.

[Response]: We thank the reviewer for the comment, and we have further polished the manuscript and checked grammar carefully. All modifications will be marked in the revised manuscript.

---

## Author Comment (AC2)

We appreciate the reviewer's recognition for our work and the valuable comments. We have followed all the comments and revised manuscript accordingly. Please check the following point-by-point responses in attachment. The reviewer's comments are in black, our responses are in blue, and the quotes from our manuscript are in italics.

[Comment]: Concentrations at a point can be affected by emissions from local as well as can be transported through long range transport. How do the authors make sure that while updating the emissions using the difference in gradients between the predicted and observed concentration, not only the local emissions but emissions from other regions having possibility of transport of concentrations to the said point are also updated?

[Response]: We thank the reviewer for the valuable comment and apologize for the lack of clarity in our manuscript. The deep learning process has considered the impact of transmission between different grids. The NN-CTM, which refers to U-Net branch in particular, employs the convolution neural network to utilize neighbor information effectively. We visualize a demo case of 3×3 convolution and 5×5 convolution in Fig. 1. In U-Net, the stacked of convolution can get the neighbor information with a bigger receptive field (e.g. stacking 5×5 convolution and 5×5 convolution can get a 9×9 convolution), the non-linear function (P-RELU) is employed to improve model fitting with nearly zero extra computational cost and little overfitting risk, and the batch normalization and dropout are employed to enhance the robustness of the model. We calculate that the receptive field of our model is 38×38 grid. In other words, the predicted pollutant concentration is related to its surrounding 38×38 grid's information, which represents the transmission between different grids. Therefore, as shown in Fig. 2, if the red gird has observation data, the surrounding blue grids' emission inventory within the receptive field will also be updated. At the same time, the grids with a longer distance will have a lower update weight. In extreme circumstances, if we have no observation data, our method will not work as we have no more information to adjust the emission inventory. If the observation data is denser, the emission inventory estimation is more accurate as it can consider more observation data.

[Figure]

Figure 1: The visualization of convolution neural network (left: 3×3 kernel size, right:

5×5 kernel size).

[Figure]

Figure 2: The visualization of neighbor emission update.

We have clarified the impact of transmission between different grids and the relation between observation data and emission inventory in the revised manuscript, as follows: (Section 2.2, Paragraph 3) *"In U-Net, the stacked of convolution can get the neighbor information with a bigger receptive field (e.g. stacking 5×5 convolution and 5×5 convolution can get a 9×9 convolution), the non-linear function (P-RELU) is employed to improve model fitting with nearly zero extra computational cost and little overfitting risk, and the batch normalization and dropout are employed to enhance the robustness of the model. We calculate that the receptive field of our model is 38×38 grid. In other words, the predicted pollutant concentration is related to its surrounding 38×38 grid's information, which represents the transmission between different grids. Meanwhile, the closer the distance, the greater the contribution."*

(Section 2.3, Paragraph 1) *"The observation data will help update the surrounding grids' emission inventory within the receptive field. However, in extreme circumstances, if we have no observation data, our method will not work as we have no more information to adjust the emission inventory. If the observation data is denser, the emission inventory estimation is more accurate as it can consider more observation data."*

[Comment]: The simulated concentrations will have some positive or negative bias based on a range of factors ranging from under/over prediction of meteorology,

chemical mechanism as well as emissions. The authors seem to add/subtract all this bias in predicted concentration to update the emission inventory. How correct is to then update the emission inventory based on biases in predicted concentration?

[Response]: We thank the reviewer for the valuable suggestion to improve the quality of the paper. There exist biases in meteorological conditions and chemical mechanism, which determines that we cannot attribute all the errors to the emission inventory. So we set constrain that the update rate of emission inventory to be a maximum of 200% compared with the prior emission for each grid when learning to ensure reasonableness. The update of the emission inventory is limited, thus leaving room for potential errors caused by other factors such as meteorology and chemical module. What's more, the updated emission inventory must be positive when learning.

On the other hand, since we are more concerned about the update of the emission inventory, we have adopted a series of methods to reduce the errors of meteorological and chemical modules as much as possible. For meteorology bias, data assimilation is adopted in WRF simulations based on observation data for the upper air and surface from National Centers for Environmental Prediction (NCEP) datasets, to ensure model performance within the benchmark range. As for chemical mechanism bias, the NN-CTM can well reproduce the CTM within an acceptable bias as introduced in Section 3.2. Such bias (<4%) is much smaller than that of simulation compared to observations which is normally more than 10% even 20%.

We have added more description for clarity in the revised manuscript, as follows:

(Section 2.3, Paragraph 3) *"Meanwhile, aiming at ensuring the reasonableness and effectiveness of estimated emission inventory, we set two constrains: 1) The update rate of emission inventory to be a maximum of 200% compared with the prior emission for each grid. There exist biases in meteorological conditions and chemical mechanism, which determines that we cannot attribute all the errors to the emission inventory. If the update ratio is very large, the NN-CTM cannot well reflect the correlation of the unseen data. Furthermore, the prior emission is accurate to a certain extent in terms of the spatial and temporal dimensions. 2)The updated emission inventory must be positive."*

[Comment]: Compared with the observed value, there are still errors in the simulation results using the new inventory, which is inevitable. Can the authors explain more about the constraints when inserting machine learning method into the emission update? In

other words, to what extent can we consider the improvement of model performance to be reasonable and acceptable.

[Response]: We thank the reviewer for the valuable comment and apologize for the lack of clarity in our manuscript. We have added more description about the constraints when learning. The update rate of emission inventory to be a maximum of 200% compared with the prior emission for each grid when learning to ensure reasonableness. The update of the emission inventory is limited, thus leaving room for potential errors caused by other factors such as meteorology and chemical module. What's more, the updated emission inventory must be positive when learning. Although we have set some constrains for emission inventory update, our method is affected by the observed data in terms of quality and sparsity. For quality, if the observed data is not accurate, the estimated emission inventory will be not meaningful. For sparsity, in extreme circumstances, if we have no observation data, our method will not work as we have no more information to adjust the emission inventory. If the observation data is denser, the emission inventory estimation is more accurate as it can consider more observation data. (Section 2.3, Paragraph 3) *"Meanwhile, aiming at ensuring the reasonableness and effectiveness of estimated emission inventory, we set two constrains: 1) The update rate of emission inventory to be a maximum of 200% compared with the prior emission for each grid. There exist biases in meteorological conditions and chemical mechanism, which determines that we cannot attribute all the errors to the emission inventory. If the update ratio is very large, the NN-CTM cannot well reflect the correlation of the unseen data. Furthermore, the prior emission is accurate to a certain extent in terms of the spatial and temporal dimensions. 2)The updated emission inventory must be positive."*

[Comment]: Page 5, Line 117. Change 'step' to 'steps'.
[Response]: We apologize for the typo, and we have modified.

[Comment]: Page 5, Line 127. Change 'the same to' to 'the same as'.
[Response]: We thank the reviewer for the suggestion and have modified.

[Comment]: Page 6, Line 140. Change 'sequence' to 'sequences'.
[Response]: We apologize for the typo and have corrected.

[Comment]: Page 6, Line 142. Remove 'a' in 'a supplementary …'.

[Response]: We apologize for the typo and have modified.

[Comment]: Page 8, Line 171. Should use 'based on the error …'.

[Response]: We apologize for the typo and have corrected.

[Comment]: Page 9, Line 199. Change 'has' to 'have'.

[Response]: We apologize for the typo and have corrected.

[Comment]: Page 9, Line 225. Notice the singular and plural forms.

[Response]: We thank the reviewer and have corrected. We also checked grammar carefully for the whole manuscript.

[Comment]: Page 11, Line 249. Change 'regions' to 'region'.

[Response]: We apologize for the typo and have modified.

[Comment]: Page 17, Line 341. Change 'achieve' to 'achieves'.

[Response]: We apologize for the typo, and we have modified.